# Understanding Soil Carbon and Phosphorus Dynamics under Grass-Legume Intercropping in a Semi-Arid Region

Amit Kumar Singh [1], Jai Bahadur Singh [1], Ramesh Singh [2], Sita Ram Kantwa [1], Prakash Kumar Jha [3,*], Safik Ahamad [1], Anand Singh [1], Avijit Ghosh [1], Mahendra Prasad [1], Shikha Singh [4], Surendra Singh [4] and P. V. Vara Prasad [3,5]

1   Indian Council of Agricultural Research—Indian Grassland and Fodder Research Institute, Jhansi 284003, India; amit.singh1@icar.gov.in (A.K.S.); jitendra.singh01@icar.gov.in (J.B.S.); safik.banda@gmail.com (S.A.)
2   International Crops Research Institute for the Semi-Arid Tropics, Hyderabad 502324, India
3   Sustainable Intensification Innovation Lab., Kansas State University, Manhattan, KS 66506, USA; vara@ksu.edu
4   Lind Dryland Research Station, Department of Crop and Soil Sciences, Washington State University, Lind, WA 99341, USA
5   Department of Agronomy, Kansas State University, Manhattan, KS 66506, USA
*   Correspondence: pjha@ksu.edu

**Abstract:** An integrated forage-legume cropping system has immense potential to address the issue of land degradability. It provides a critical understanding of the capacity of diversified species mixes vs. monocultures to boost forage production and the dynamics of soil organic carbon (SOC) and phosphorus (P). In this study, we assessed the performance of Napier Bajra Hybrid (NBH) (*Pennisetum glaucum* × *P. purpureum*) + cowpea (*Vigna unguiculata*) and tri-specific hybrid (TSH) (*P. glaucum* × *P. purpureum* × *P. squamulatum*) + cluster bean (*Cyamopsis tetragonoloba*) as compared to monocultures of NBH and TSH. The legume equivalent yield of NBH + cowpea and TSH + cluster bean intercropping systems were found −31% and −23% higher than monoculture systems. The SOC increased by −5% in the NBH + cowpea system as compared to NBH monoculture. The carbon mineralization rates under NBH + cowpea and TSH + cluster bean were −32% and −38% lower than the NBH and TSH monoculture cropping systems, respectively. It was found that the legume intensification with the forage significantly improved the soil's P status. The research suggested that coalescing diverse crops (e.g., grass and legume) poses enormous potential for sustaining soil health and productivity in semi-arid regions of India. This study advances the research on characterizing the crucial factors of grass-legume-based cropping systems and helps in assessing the impact of these factors on long-term sustainability.

**Keywords:** legume intensification; soil organic carbon pool; soil phosphorus fractions; forage yield

## 1. Introduction

According to current estimates, the world population is expected to grow beyond nine billion by the end of the year 2050, adding more than two billion people to the current population [1]. An increased population will entail further exploitation of natural resources, which necessitates the sustainable use of natural resources, including agricultural production systems [2]. The increasing population pressure and its repercussions led to the production of more food crops, hence diverting farmers' attention away from forage/fodder crop cultivation. The deficiency of nutritious fodder crops negatively impacts cattle productivity. Therefore, choosing effective land use decisions, such as crop diversification and intensification, will ensure the better utilization of resources [3]. Crop productivity enhancements using crop diversification are considered one such means to achieve these long-term goals. In contrast to monoculture, crop diversification aims to expand the diversity of

crops using methods such as crop rotation, multiple cropping, or intercropping to promote production, stability, and the provision of ecosystem services [4,5]. It is a step towards creating more environmentally friendly agricultural methods, value chains for minor crops, and socioeconomic advantages [6]. Intercropping perennial fodder crops with legumes is one such popular practice in semi-arid India because of greater land use efficiency and improved soil fertility due to nitrogen fixation [7]. Integration of perennial fodder crops with fodder legumes is considered a potential solution to this problem. The study of the impact of the perennial fodder-based system on soil health will support the sustainable use pattern of resources [8]. Legume crops can increase soil organic matter (SOM), which helps improve nutrient cycling and availability, erosion control, soil-water movement and retention, soil conservation, soil biota, and buffering limitations [9]. Furthermore, legume residues produce more SOM, affecting soil aggregation and lowering soil bulk density. The breakdown of SOM provides plants with readily available nutrients [10]. The soil organic carbon (SOC) concentration in soil is considered the primary indicator of soil health and SOC pools, such as mineral-associated organic carbon, particulate organic matter-associated carbon, and active C ($KMnO_4$), show the status of organic carbon concentrations in the soil, hence providing resilience to the soil against the climate change [11].

Studies have shown that inter-species intercropping produces higher yields than intra-species intercropping, which uses various cultivars of the same species [12]. The capacity of intercrops to mobilize soil P is also influenced by the availability of phosphorus. The $NaHCO_3$-P concentration has been used as a useful indicator of soil P availability [13]. However, $NaHCO_3$-P in the soil is not a single entity, a quantitative analysis of soil P accumulation into various pools, such as saloid-P, aluminum phosphate (Al-P), iron phosphate (Fe-P), reduction phosphate (Red-P), and calcium phosphate (Ca-P) are more relevant for predicting the soil P status change in grass-legume intercropping systems. In degraded soil, C dynamics and P transformation are largely controlled by microbial activity. Hence, estimating soil microbial activity, such as urease, alkaline phosphatase, dehydrogenase, and glucosidase, would depict the true status of the elemental interactions in grass-legume systems. The impact of grass-legume intercropping on the activities of soil enzymes has rarely been studied in degraded land [14] and is needed. In addition, knowledge is scarce about the role of grass-legume intercropping on the availability and interaction of nutrients in semi-arid degraded land. Napier-bajra hybrid (NBH) and *Pennisetum* tri-specific hybrids are the most widely adopted perennial fodder crops in India. They are known for their rapid resurgence and high biomass production capacity. Cowpeas (*Vigna unguiculata*) and cluster beans (*Cyamopsis tetragonoloba*) are two important leguminous crops suitable for semi-arid India. These legumes are also used as fodder crops. However, intercropping of grasses and legumes aid to supply a balanced diet to animals, supplying carbohydrate and protein simultaneously. Intercropping could also sustain soil health.

Therefore, this study was conducted to assess whether legume-based crop diversification will sustain C and P dynamics or not. We hypothesized that plants in mixtures would grow better than those in monocultures, because of N fixation and P mobilization by legumes.

## 2. Materials and Methods

### 2.1. Experimental Site

The research was conducted at the institutional farm of the Indian Council of Agricultural Research—Indian Grassland and Fodder Research Institute (ICAR—IGFRI), Jhansi (25.51° N 78.53° E), located in the Bundelkhand region of India. The climate of the region is characterized by erratic rainfall and frequent droughts. The soil of the region has poor nutrient supply capacity and deteriorating health. The region depends on the summer monsoon for precipitation, which usually extends from the end of June to mid-September. The summer monsoon in the region is responsible for 85–90% of rainfall and the remaining rainfall is usually the manifestation of western disturbances in North India from January to April. The average annual precipitation of the Jhansi region varies between 820–840 mm. May and January are the hottest and coolest months, respectively, of the year, having a

median daily warmth of 42 °C and 5 °C. The soil at the test site is categorized as typic Haplustepts (fine-loamy, mixed, hyperthermic typic) [15]. There are 366 g kg$^{-1}$ of sand, 432 g kg$^{-1}$ of silt, and 202 g kg$^{-1}$ of clay in the soil. The soil had a pH of 7.29, 212 kg of mineral N ha$^{-1}$, 12.03 kg of extractable P, and 225 kg of exchangeable K ha$^{-1}$ in the top 15 cm of soil. At soil depths of 0–15 cm and 15–30 cm, respectively, the initial soil organic carbon content was 3.8 and 3.6 g kg$^{-1}$. The soil's saturation water capacity and field capacity was 59.5 and 11.8% (*v/v*), respectively.

### 2.2. Establishment of the System

The experiment was laid out in a randomized block design (RBD) from the rainy season of 2016 and continued until 2021 in the central research farm of ICAR—IGFRI. Four cropping systems, namely including perennial fodder grasses such as Napier-bajra hybrid (NBH) and *Pennisetum* tri-specific hybrid (TSH), NBH + cowpea, and TSH + cluster bean, were established in 9 × 3 m plots. All systems were replicated thrice, and treatments were randomized using a random number table. The row-to-row distance for NBH and TSH was maintained at 100 cm, while the plant-to-plant distance was maintained at 50 cm. Annual crops of cowpea and cluster beans were sown, maintaining a row-to-row distance of 30 cm and plant-to-plant distance of 10 cm. The NBH and TSH were planted in the month of July 2016, while cowpea and cluster beans were sown in the interspaces in two rows in the same season in 2016. A standard package of practices was followed for all crops as available in the literature and presented in Table S1. Around 60 to 75 days after sowing, all crops were harvested and the yield of green fodder was calculated. Randomly selected samples of chopped green fodder were sun-dried and then dried for 72 h at 65 °C to determine the dry matter percentage, which was then multiplied by the appropriate green fodder yield to determine the dry matter yield. These yields were converted into legume-equivalent yields for each system by multiplying the grass yield with the ratio of the grass market price to the legume component market price [16].

### 2.3. Soil Sample Collection, Processing, and Analysis

Soil samples were collected from the surface (0–15 cm) and sub-surface soil (15–30 cm) at the beginning of the experiment in July 2016, in triplicates, and again in April 2021 at the end of the harvesting season for soil chemical and enzymatic analysis from each plot. Each sample was divided into two subsamples. The first subsample was dried, processed, and put through a 4.75-mm screen (bulk soil) to estimate the soil's chemical characteristics. For the evaluation of soil enzymatic activity, the second subsample was stored in a refrigerator at the ideal temperature (4 °C).

### 2.4. Soil Analysis

Deionized water was used to measure the pH of the soil (1:2.5 soil/water). The salinity of the soil was assessed using the electrical conductivity of an aqueous soil extract in deionized water. N, P, S, and K were expected to be accessible to plants [17]. The Walkley–Black technique was used to calculate the SOC [18]. The water-soluble carbon (WSC) amount was calculated using the hot water extraction technique [19]. By using modified wet chromate oxidation with 18 N $H_2SO_4$, labile (LC) and recalcitrant carbon (RC) were determined [20]. The approach of [21] was used to determine the amount of mineral organic matter-associated carbon (MOM-C) and particulate organic matter-associated carbon (POM-C). The following formulae were used to determine the active C and the C management index [22].

$$Lability\ of\ C_L = \frac{C_L}{C_{NL}} \tag{1}$$

$C_L$ is the C fraction oxidized by $KMnO_4$ and $C_{NL}$ is the C remaining unoxidized by $KMnO_4$.

$$Carbon\ pool\ index\ (CPI) = \frac{TOCer}{TOCr} \tag{2}$$

*TOCer* is the soil organic carbon in the harvested field and *TOCr* is soil organic C in the initial soil.

$$Carbon\ lability\ index\ (LI) = \frac{Ler}{Lr} \tag{3}$$

*Ler* is the lability of carbon in the harvested field and *Lr* is the lability of carbon in the initial soil.

$$Carbon\ management\ index(CMI) = CPI \times LI \times 100 \tag{4}$$

For 31 days, the kinetics of carbon decomposition in bulk soils were studied at 250 °C in incubators. While 15 days were spent pre-incubating the soil samples at 240 °C and 75% of the soil field capacity [13]. Using a pressure plate apparatus, the field capacity values for various treatments were calculated at a water potential of −30 kPa. At each sample date, the evolved $CO_2$ was measured after being captured by 10 mL of 0.5 N NaOH (in the alkali trap) (days 2, 4, 7, 10, 17, and 24). The quantities of $CO_2$ captured were measured by back titrating 0.5 N NaOH with 0.5 M HCl at a pH of 8.3 in the presence of $BaCl_2$ in order to compute the C mineralization rates. After removing the NaOH on each sample date, deionized water was added to the flasks to maximize moisture (75% of the soil field capacity), and compressed air was pumped into the containers to enable $O_2$ delivery. The formula used to calculate the $CO_2$ flow (mg kg$^{-1}$) was:

$$CO_2 - C\ evolved = (A - B) \times N \times 6 \tag{5}$$

where *N* is the normality of HCl, 6 is the equivalent weight of *C*, and *A* and *B* are the volume (mL) of HCl needed to titrate 10 mL of 0.5 M NaOH in the control (flask without soil) and soil, respectively. To calculate *C* loss over time, an exponential model [23] was used.

$$Ct = Co(1 - \exp(-k \times t)) \tag{6}$$

where *Co* represents the potentially mineralizable C, and *Ct* is the pool of C mineralized at time (*t*), with decay rate (*k*).

Phosphorus fractions, such as saloid P (sal-P), aluminum (Al-P), iron (Fe-P), calcium-bound P (Ca-P), organic P (org-P), and reductant soluble P (Res-P), were estimated using [24] methodology. Dehydrogenase and β-D-glucosidase, two carbon cycling enzymes, were evaluated using triphenyl tetrazolium chloride and p-nitro phenyl-D-glucopyranoside [25]. Urea and p-nitro phenyl phosphate were used as the substrates for the evaluation of N and P cycling enzymes such as urease and alkali phosphatase [25].

### 2.5. Soil Enzyme Activity as Indicator

2.5.1. Soil Quality Index

For each sample, the tested enzymes' geometric mean enzyme activity (GMEA) was estimated as follows:

$$GMEA = (DHA \times GLU \times ALP \times URE)^{\frac{1}{4}} \tag{7}$$

where *DHA*, *GLU*, *ALP*, and *URE* stand for dehydrogenase, β-D-glucosidase, alkali phosphatase, and urease activity, respectively. The *GMEA* integrates the multiple enzyme activities to represent the soil functions, hence, it is used as a proxy for the soil quality index [26].

In accordance with [27], an enzyme-based treated soil quality index was generated.

$$T - SQI = 10^{\log m + \frac{\sum_{i=1}^{n}(\log n_i - \log m) - \sum_{i=1}^{n}(\log n_i - \log ň)}{n}} \tag{8}$$

where *m* is the reference (set to 100%) and *n* is the measured percent of the reference, *ň* indicates the mean values of soil enzymes.

2.5.2. Soil Functional Diversity [28,29]

It was determined using Shannon's Diversity Index (*H*) [30]

$$H = -\sum_{i=1}^{4} P_i \times \ln(P_i) \tag{9}$$

and the Simpson–Yule Index (*SYI*) [31].

$$SYI = \frac{1}{\sum_{i=1}^{4} P_i^2} \tag{10}$$

where $P_i$ is the ratio of each enzyme activity to the sum of all enzyme's activities for a particular sample. In all cases, enzyme activities were expressed as µg product formed per g of soil per hour.

*2.6. Statistical Analysis*

SAS 9.3 was used to perform a statistical analysis (Cary, NC, USA). When the F-test revealed that there were factorial effects with a significance threshold of $p < 0.05$, a one-way analysis of variance was conducted, and the means were separated by least significant differences (LSD). Moreover, the regression analysis was performed to understand the impact of weather parameters on yield and soil P dynamics.

**3. Results**

*3.1. Weather Parameters during the Study Period*

The year 2018 had the most rainfall during the *Kharif* season (July to October), with 775 mm, followed by 660 mm in 2019 and 455 mm in 2017. In all of the studied years, the rainfall that was received in three months (July to September) was between 75 and 90 percent of the total annual rainfall. Throughout the research period, the *Rabi* season's (15–40 mm) rainfall was quite low. The average maximum and minimum monthly temperatures and average relative humidity were nearly constant over all the years (Figures 1 and S1).

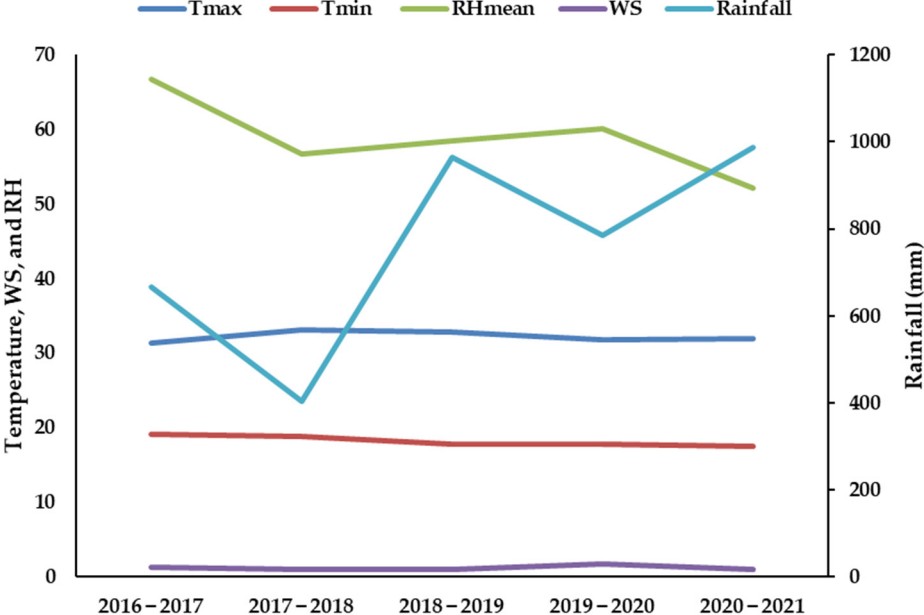

**Figure 1.** Weather parameters (maximum temperature (Tmax; °C), minimum temperature (Tmin; °C); relative humidity (RH) mean (RHmean; %), wind speed (WS; km h$^{-1}$), and rainfall (mm) during the study period (2016–2021).

### 3.2. Soil Properties

Nearly a 25 and 14% increase was observed for the NBH + cowpea intercropping system in the surface layer of the soil for the available N and S, respectively, in comparison to monocultures. However, no significant differences were observed for the available P and K in comparison to the monoculture NBH grass. In the TSH + cluster bean intercropping system, the available N and K increased by −15 and −13% over the TSH monoculture grass system, respectively (Table 1). In the subsurface soil of the NBH + cowpea intercropping system, a −12 and −11% increase was observed for the available N and K, respectively, but the available P decreased by −5%. No significant difference was found for the available S in comparison to the monoculture NBH grass. In the TSH + cluster bean intercropping system, the available N and K showed a corresponding increase of −20 and −34% over the TSH monoculture grass system. However, the available S decreased by −19%.

**Table 1.** Impact of different cropping systems on soil mineral N (kg ha$^{-1}$), available P (kg ha$^{-1}$), K (kg ha$^{-1}$), and S (kg ha$^{-1}$) in 0–15 and 15–30 cm soil layers in a semi-arid Inceptisol.

| Cropping System | Legume Equivalent Yield | Dry Fodder Yield | Mineral N | Available P | Available K | Available S |
|---|---|---|---|---|---|---|
| | | | **0–15 cm** | | | |
| NBH + cowpea | 16.93 a | 16.61 a | 156.80 a | 12.39 a | 82.13 a | 39.94 a |
| TSH + cluster bean | 15.76 b | 15.1 b | 131.71 b | 12.60 a | 64.59 b | 32.42 b |
| NBH monoculture grass | 12.94 c | 12.23 c | 125.44 b | 12.68 a | 86.24 a | 34.84 b |
| TSH monoculture grass | 12.77 c | 12.16 c | 114.43 c | 12.10 ab | 57.12 c | 32.69 b |
| | | | **15–30 cm** | | | |
| NBH + cowpea | 16.93 a | 16.61 a | 137.98b | 12.17 b | 84.00 a | 39.94 a |
| TSH + cluster bean | 15.76 b | 15.1 b | 150.53a | 13.04 a | 81.01 b | 33.77 b |
| NBH monoculture grass | 12.94 c | 12.23 c | 123.07c | 12.89 a | 75.41 b | 40.21 a |
| TSH monoculture grass | 12.77 c | 12.16 c | 125.44 c | 12.97 a | 60.48 c | 41.82 a |

Means with similar lower-case letters within a column are not significantly different as per LSD ($p < 0.05$).

### 3.3. Crop Productivity

The legume equivalent yield of the NBH + cowpea system was found to be the highest among all the systems. This system has −7% higher yields over the intercropping system of TSH + cluster bean over the experimental period of five years. In comparison to the monoculture NBH and TSH system, a legume equivalent yield of the NBH + cowpea system was found 31 and 23% higher in comparison to the monoculture crops (Table 1).

### 3.4. SOC and Its Fractions

In the surface soil, the organic carbon increased by −5% in the NBH + cowpea system in comparison to the NBH monoculture grass, however, no significant difference was observed for the TSH + cluster bean system in comparison to the TSH monoculture grass for the surface soil (Table 2). In the NBH + cowpea intercropping system, a −70, −14, and −4% increase was observed for the labile carbon, KMnO$_4$-C, and POM-C, respectively, in 0–15 cm soil depth in comparison to the monoculture NBH grass. However, for the RC and MOM-C pools, a −100 and −11% decrease was observed in the NBH + cowpea intercropping system, respectively (Table 2). In the TSH + cluster bean intercropping system, a −54 and −17 % increment was observed for LC and KmnO$_4$-C, respectively, while RC and MOM-C decreased by −47 and −6%, respectively, in comparison to the TSH monoculture grass system. There was no significant difference observed for the POM-C in similar system interventions, however, among the dissimilar set of systems, a difference was significant. No significant difference was found for WSC among the four systems (Table 2).

**Table 2.** Impact of different cropping systems on soil organic carbon (SOC; g kg$^{-1}$), labile C (LC; g kg$^{-1}$), recalcitrant C (RC; g kg$^{-1}$), water-soluble C (WSC; mg kg$^{-1}$), KMnO$_4$-C (AC; mg kg$^{-1}$), particulate organic matter C (POM; g kg$^{-1}$), mineral organic matter C (MOM; g kg$^{-1}$), and carbon management index (CMI) in 0–15 and 15–30 cm soil layers in a semi-arid Inceptisol.

| Cropping System | SOC | LC | RC | WSC | KMnO$_4$-C | POM | MOM | CMI |
|---|---|---|---|---|---|---|---|---|
| | | | | **0–15 cm** | | | | |
| NBH + cowpea | 5.6 a | 4.41 a | 1.14 d | 87.00 a | 3.92 a | 0.22 b | 1.11 b | 85.48 a |
| TSH + cluster bean | 5.2 b | 3.82 b | 1.33 c | 89.50 a | 4.09 a | 0.25 a | 0.90 c | 89.14 a |
| NBH monoculture grass | 5.1 b | 2.82 c | 2.28 b | 85.00 a | 3.42 b | 0.21 b | 1.25 a | 74.65 b |
| TSH monoculture grass | 5.0 b | 2.47 d | 2.53 a | 87.50 a | 3.49 b | 0.25 a | 0.96 c | 76.16 b |
| | | | | **15–30 cm** | | | | |
| NBH + cowpea | 4.8 a | 4.01 a | 0.74 d | 88.00 a | 4.05 c | 0.26 a | 1.47 a | 82.12 c |
| TSH + cluster bean | 4.4 b | 3.02 b | 1.33 c | 91.50 a | 4.46 b | 0.19 b | 1.19 c | 90.48 ab |
| NBH monoculture grass | 4.7 a | 2.62 c | 2.08 a | 85.50 a | 4.23 c | 0.17 c | 1.25 b | 85.94 bc |
| TSH monoculture grass | 4.2 b | 2.42 c | 1.78 b | 85.50 a | 4.61 a | 0.17 c | 1.13 c | 93.49 a |

Means with similar lower-case letters within a column are not significantly different as per LSD ($p < 0.05$).

For the subsurface soil, no significant difference was observed for the SOC in either of the intercropping systems (Table 2). The NBH + cowpea intercropping system SOC, LC, POM-C, and MOM-C showed an increase of −2, −53, −52, and −18% in comparison to the monoculture NBH grass, respectively. However, RC and KMnO$_4$-C decreased by −64 and −4% in the subsurface soil, respectively (Table 2). The TSH + cluster bean intercropping system enhanced LC, POM-C, and MOM-C by −25, −12, and −5% in comparison to the monoculture TSH grass, respectively. RC and KMnO$_4$-C revealed a decrease of −25 and −3%, respectively. No significant difference was found for WSC among the four systems (Table 2). The CMI for the surface soil increased by −15 and 17% for the NBH + cowpea and TSH + cluster bean intercropping systems in comparison to the NBH monoculture and TSH monoculture, respectively. However, this trend was found contrary for the subsurface soil i.e., decreasing for the intercropping systems (Table 2).

*3.5. Carbon Decay Kinetics*

The SOC carbon mineralization rate was significantly reduced by legume incorporation in both soil layers. The carbon mineralization rates under the NBH + cowpea and TSH + cluster bean were −32 and 38% lower than the NBH and TSH monoculture cropping, respectively. A similar trend was true for the next soil layer also. Potentially mineralizable C also decreased significantly due to the legume intensification in both soil layers (Table 3).

**Table 3.** Impact of different cropping systems on soil organic carbon decay rates (K$_c$; per day) and potentially mineralizable C (C$_0$, ppm) in 0–15 and 15–30 cm soil layers in a semi-arid Inceptisol.

| Cropping System | 0–15 cm | | 15–30 cm | |
|---|---|---|---|---|
| | K$_c$ | C$_0$ | K$_c$ | C$_0$ |
| NBH + cowpea | $1.75 \times 10^{-2}$ c | 53.90 b | $2.29 \times 10^{-2}$ b | 38.60 c |
| TSH + cluster bean | $2.45 \times 10^{-2}$ b | 57.60 b | $2.11 \times 10^{-2}$ b | 47.80 c |
| NBH monoculture | $2.56 \times 10^{-2}$ b | 53.60 b | $3.71 \times 10^{-2}$ a | 85.60 b |
| TSH monoculture | $3.98 \times 10^{-2}$ a | 94.40 a | $3.38 \times 10^{-2}$ a | 112.00 a |

Means with similar lower-case letters within a column are not significantly different as per LSD ($p < 0.05$).

*3.6. Soil P Fractions*

In the surface layer, saloid-P decreased by −25 and −29% in the NBH + cowpea and TSH + cluster bean intercropping systems in comparison to the monoculture NBH and TSH, respectively (Figure 2). Al-P decreased by −14 and 5% in the NBH + cowpea and TSH + cluster bean intercropping systems in comparison to the monoculture NBH and



TSH, respectively (Figure 2). Nearly a −15% increase was observed in the NBH + cowpea intercropping for Fe-P for a 0–15 cm soil depth in comparison to the monoculture NBH grass. However, no significant difference was observed in the Fe-P pool of phosphorus in the TSH + cluster bean intercropping system in comparison to the monoculture TSH System (Figure 2). No significant difference was observed in the Ca-P in the NBH + cowpea intercropping system and monoculture NBH, however, the TSH + cluster bean intercropping system exhibited a −40% increase in the Ca-P pool in comparison to the monoculture TSH (Figure 2). No significant difference was observed in the total P and Res-P pools of phosphorus in both the NBH + cowpea and TSH + cluster bean intercropping systems in comparison to the monoculture crop (Figure 1). Nearly an 18 % increase was observed in the NBH + cowpea intercropping for organic-P for a 0–15 cm soil depth in comparison to the monoculture NBH grass, however, no significant difference was observed in the TSH + Cluster bean intercropping system (Figure 2).

The subsurface soil saloid-P decreased by −29% in the NBH + cowpea system over the monoculture NBH, however, it increased by −10% in the TSH + cluster bean intercropping system over the monoculture TSH (Figure 2). The Al-P decreased by −11% in the NBH + cowpea intercropping system over the monoculture NBH. However, no significant difference was observed in the TSH + cluster bean intercropping system in comparison to the monoculture TSH System (Figure 2; Table S2). No significant difference was observed for the Fe-P in NBH + cowpea and TSH + cluster bean intercropping systems in comparison to their monoculture systems (Figure 2). The Ca-P increased by −124% in the NBH + cowpea intercropping system over the monoculture NBH, however, it decreased by −34% in the TSH + cluster bean intercropping system over the monoculture TSH. No significant difference was observed among the total P in the intercropping systems and the monoculture crops (Figure 2). The Res-P pool was increased by −19% in the NBH + cowpea system, however, no significant difference was observed in the TSH + cluster bean intercropping system.

### 3.7. Soil Quality Index and Functional Diversity Index

In the NBH + cow pea and TSH + cluster bean intercropping systems, GMEa was found to be −55 and 30% higher in comparison to the monoculture NBH and TSH, respectively, in the surface soil (Table 4). Shannon's Diversity Index (H) showed non-significant improvement in the surface soil, however, a significant improvement was found in the subsurface soil (Table 4). The T-SQI values were found to be significantly higher in the surface soil. This was -165 and 122% higher for the NBH + cowpea and TSH + cluster bean systems over the monoculture NBH and TSH, respectively. For the subsurface soil, the T-SQI values were −111 and 81% higher for the NBH + cowpea and TSH + cluster bean systems over the monoculture NBH and TSH, respectively (Table 4).

**Table 4.** Impact of legume intensification on GMEa, Shannon's Diversity Index (H), Simpson–Yule Index (SYI), and T-SQI in 0–15 and 15–30 cm soil layers in a semi-arid Inceptisol.

| Cropping System | GMEa | H | SYI | T-SQI |
|---|---|---|---|---|
| | | **0–15 cm** | | |
| NBH + cowpea | 12.48 a | 0.77 a | 1.85 a | 265.12 a |
| TSH + cluster bean | 9.84 b | 0.72 b | 1.74 b | 221.59 b |
| NBH monoculture | 8.03 c | 0.75 ab | 1.83 ab | 175.82 c |
| TSH monoculture | 7.60 c | 0.68 b | 1.66 b | 179.21 c |
| | | **15–30 cm** | | |
| NBH + cowpea | 8.25 a | 0.86 a | 2.13 a | 210.58 a |
| TSH + cluster bean | 7.92 a | 0.70 b | 1.70 b | 180.55 b |
| NBH monoculture | 4.58 b | 0.56 c | 1.46 c | 147.66 c |
| TSH monoculture | 4.04 c | 0.43 d | 1.29 d | 144.75 c |

Means with similar lower-case letters within a column are not significantly different as per LSD ($p < 0.05$).

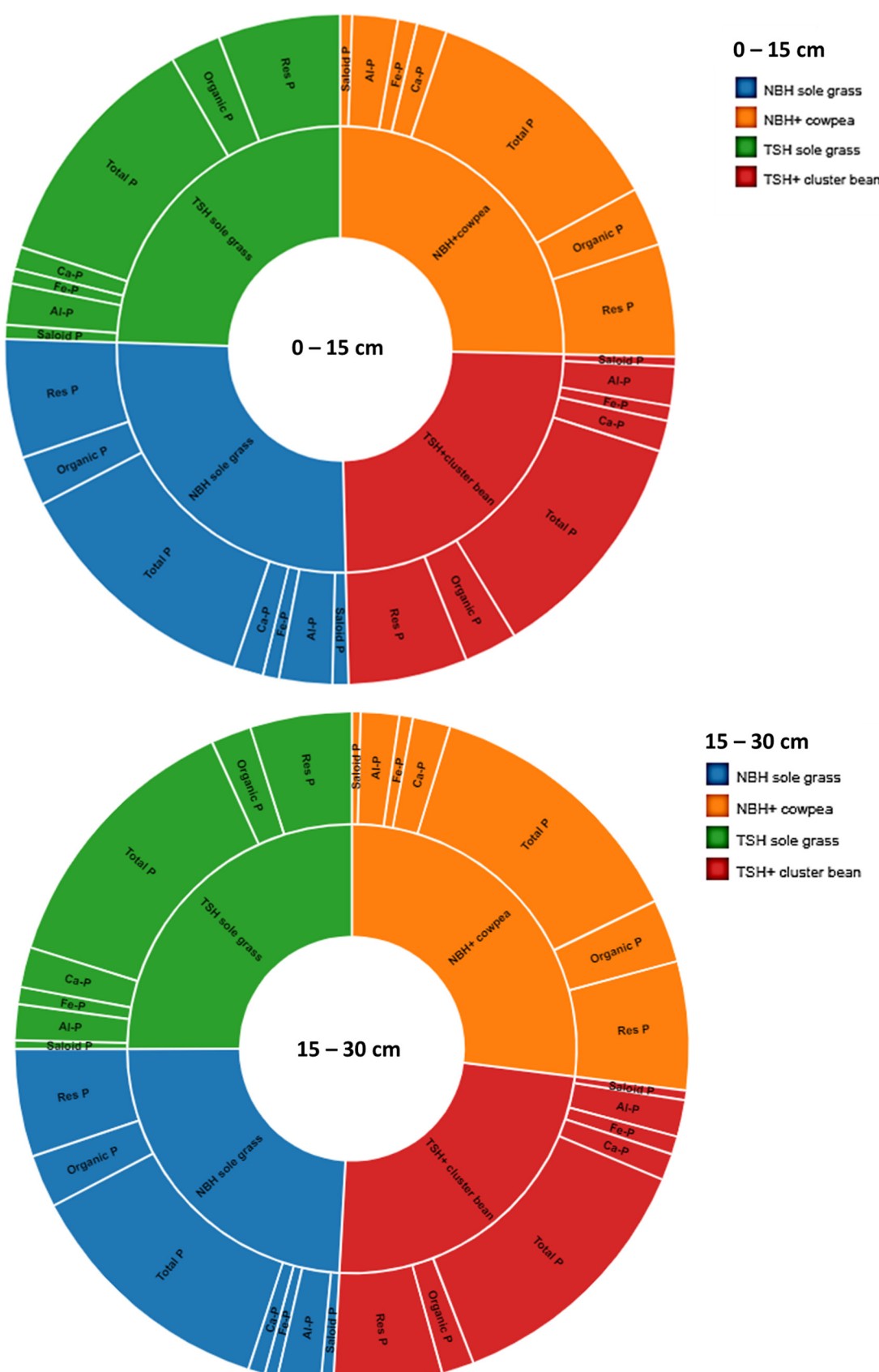

**Figure 2.** Impact of different cropping systems on saloid P, aluminum, iron, calcium bound P, organic P, and reductant soluble P in 0–15 and 15–30 cm soil layers in a semi-arid Inceptisol.

## 4. Discussion

### 4.1. Crop and System Productivity

Legume intensification also ensured a better soil P and N status (Table 2). With climate change, variations in average temperatures and precipitation occur, which speed up soil organic matter breakdown or erosion rates [32,33]. However, legume intensification is ascertained to be capable of slowing soil nutrient loss as observed from a leachate analysis through its proper utilization. This could be observed from the lower concentration of Na, K, carbonate, and Ca + Mg in the leachate samples collected from different cropping systems (Table S3). Legume incorporation also confirmed a greater availability of moisture and nutrients and greater root biomass of the intercropping systems (Table S3). The regression analysis revealed that climatic parameters, such as rainfall, maximum, and minimum temperature, were impacting the yield of the monoculture and legume-intensive cropping systems ($R^2 > 0.723$; $p < 0.05$). Despite that, the legume-intensive cropping systems resulted in higher yields (Table 1). Thus, even when environmental conditions were influencing the system, legume intensification strategies created a resilient soil environment for optimal plant development [15]. Hence, improved soil moisture retention enhanced the SOC content with better nutrient recycling, and higher outputs were obtained under the legume-intensified systems. Consequently, the legume equivalent yield was higher for them (Table 1). Crop growth boosted SOC and microbial activity while increasing crop production. The cycle repeats to amplify the ecological services [3]. The economic analysis revealed that the benefit-cost ratio followed the order NBH + cowpea (2.47:1) > TSH + cluster bean (1.96:1) > cowpea (1.83:1) > Guar (1.67:1).

### 4.2. Soil Organic Carbon Pools

The active and passive pools of SOC were significantly affected by various cropping sequences across various depths (Table 2). In general, the active C pools (LC, $KMnO_4$-C) were higher under the legume-intensified systems, whereas RC was lower under those systems as compared to the monoculture grass-based cropping systems. This may be an indication that legumes have used the active carbon pool more significantly and prevented carbon losses from the soil (Table 2). Leguminous crop cultivation had been linked to higher SOC levels in deteriorated soil of subtropical climates [11,13,34]. The atmospheric C is taken up by plant roots, transferred into the soil in the form of C-containing compounds, and then stored there for extended periods as AC and RC. Significant amounts of carbon are deposited in different soil levels thanks to root exudates and lysates, although the quantity of carbon deposited in the soil layers varies according to the crop [31]. We speculate that these variables might be the main ones influencing the distribution and wide range of trends of the active and recalcitrant C pools. Thus, legumes-based cropping intensification for enhancing both pools of C in the soil could be suggested.

Crop diversification/intensification often results in a greater C pool than monotonous sequences under a variety of management strategies [35]. Engineering for cropping sequences encouraged the SOC status since SOC existed in the soil's uppermost layers [36]. Compared to the monocropping, the legumes' low C-N ratio may have been an extra benefit that considerably increased the stability of labile C [37]. In many cases, growing high biomass-generating fodder crops in intensive cropping cycles led to higher soil C levels [38]. The study's findings will help researchers better understand the varying effects of crop intensification on soil C content in a system for producing fodder.

The balance between C inputs and losses leads to the formation of soil carbon pools. In the current investigation, intensified cropping systems may have affected SOC storage through a variety of mechanisms, including their impact on SOC decomposition kinetics and reduction in C decay rates (Table 3), since these systems offered virtually year-round soil cover, increasing the net annual C intake. When compared to monoculture cropping, fields with legume intensifications had a higher SOC and higher MOM and POM fractions while maintaining high rates of productivity, indicating that increased inputs from intensification and easily metabolizable inputs from legumes support SOC formation to sustain

productivity through POM turnover and confer long-term persistence as MOM. This shows how important it is to not only boost production to encourage increased SOC but also how the inclusion of certain crops in the intensification process may greatly affect the stability and the capacity to acquire SOC. Decreased C-N ratios of legume wastes improved soil aggregation, etc., may all contribute to lower C decay rates in the legume intensification systems [34]. Legumes have higher sequestration effectiveness than monoculture cropping, as seen by the decrease in potentially mineralizable C owing to the intensification of the crop.

### 4.3. Soil Phosphorus Fractions

Our study found a benefit in P uptake from the soil P for the NBH + cowpea crop intensification. This study demonstrates how the cropping strategy has a significant impact on soil P fractions when the cultivation time is increased. It is not unexpected that the most labile P (saloid P) was reduced by various cropping systems (monoculture crops and legume intensification systems). However, similar to other studies, the depletion in other P fractions varied significantly with crop attributes [39,40]. Since legumes may exude root exudates that increase the P availability, they primarily employ saloid P and Al-P (Figure 2) [39,40]. The crop is unable to exploit the acid-soluble inorganic P pool and relies heavily on organic P, while NBH and TSH's potential to change the rhizosphere is comparatively limited (Figure 2) [39,40]. This may be because solitary grasses were cultivated on infertile soil and arbuscular mycorrhizas were crucial for P uptake [39,40]. The capacity of arbuscular mycorrhizal fungi and associated hyphae microbiomes to promote soil organic P mineralization under field conditions was linked to the function of the bacterial community on the hyphae surface [41]. Future studies must be conducted to ascertain if the differences in organic P depletion between single and legume-enhanced grass production systems demonstrate the distinct roles of released chemicals and microbial activity in organic P mineralization. Thus, solitary grasses contributed to soil P depletion through the use of saloid P and organic P (Figure 2). The elimination of intercropped legume-modified soil fractions, through time, is likely what brought about the identical condition of Fe and Ca-bound P. To avoid having divergent results between short-term greenhouse experiments and long-term field trials, the intercropping arrangement must be taken into consideration when examining the P fraction variations in a long-term field experiment. Further research is required in this field of study.

### 4.4. Activities of Soil Enzymes

Increased labile soil carbon in those plots may be the cause of the much higher DHA activity for the legume intensification systems compared to the solo crops. The legume intensification systems had a much greater alkaline phosphatase activity than the solitary crops. Dodor and Tabatabai [42] assert that SOC can control the phosphatase activity of the soil. In the current study, a significant linear relationship between SOC and soil phosphatase activity was observed ($R^2 = 0.812$; $p < 0.05$). Because of this, agroecosystems obtaining organic C, N, P, and S from diverse sources (such as grass and legumes) may affect the microbial transformation of organic matter and the functional diversity of the microbial population in the soil. Soil functional diversity is controlled by the amount, quality, and microbiological accessibility of the substrate [43]. Organic and inorganic nutrients have different pathways for microbial uptake and breakdown. Due to this, plots with a greater concentration of legumes than the mono-crops may have a higher Shannon's Diversity Score. In addition to improving soil properties, legumes, such as cowpeas, can also be used as a staple food, such as in semi-arid regions of Brazil [44], hence, they can contribute to food security also.

### 5. Conclusions

This long-term study was conducted by the incorporation of legumes (cowpea and cluster bean) in the popular NBH and TSH perennial forage-based system. The study

highlights the role of legume intensification in fodder cultivation systems to improve the yield and C and P status of the soil. The results indicated that the NBH + cowpea and TSH + cluster bean cropping intensification significantly improved the yield, as well as the C and P fractions for the surface and subsurface soils. They also helped in reducing the C mineralization rates and potentially mineralizable C, implying a better sequestration efficiency. We also found that NBH + cowpea cropping intensification had an advantage in P acquisition from the soil P, implying lesser eutrophication potential. Soil enzymatic activity and functional diversity also significantly improved under the NBH + cowpea cropping system. Therefore, the NBH + cowpea system could be recommended in degraded soils of the central semi-arid region of India.

**Supplementary Materials:** The following supporting information can be downloaded at: https://www.mdpi.com/article/10.3390/agronomy13071692/s1, Table S1: Package of Practices of NBH, TSH, Cowpea and Cluster bean crops. Table S2: Impact of legume intensification on saloid P, aluminium, iron, calcium bound P, organic P and reductant soluble P (ppm) in 0–15 and 15–30 cm soil layers in a semi-arid Inceptisol. Means with similar lower-case letters within a column are not significantly different as per LSD ($p < 0.05$). Table S3: Impact of legume intensification on carbonate, calcium + magnesium, pH, water soluble C (WSC), potassium, phosphorus, and sodium concentration in leachates in a semi-arid Inceptisol. Figure S1: Yearly Crop Evapotranpiration (mm) for NBH, TSH, Cowpea and Cluster bean crops.

**Author Contributions:** Conceptualization, A.K.S. and J.B.S.; methodology, A.K.S. and A.G.; software, R.S.; validation, S.R.K., S.A. and A.S.; formal analysis, A.K.S.; investigation, A.G. and M.P.; resources, J.B.S. and S.R.K.; data curation, A.K.S. and A.G.; writing—original draft preparation, P.K.J. and A.K.S.; writing—review and editing, S.S. (Shikha Singh), A.G. and P.V.V.P.; visualization, P.K.J., A.K.S., S.S. (Surendra Singh) and P.V.V.P.; supervision, J.B.S. and A.K.S.; project administration, J.B.S., A.K.S. and S.R.K.; funding acquisition, J.B.S. All authors have read and agreed to the published version of the manuscript.

**Funding:** This research received no external funding.

**Data Availability Statement:** The data and materials will be made available from the corresponding author(s) upon reasonable request.

**Acknowledgments:** All the authors acknowledge ICAR-Indian Grassland and Fodder Research Institute, Jhansi 284003, Uttar Pradesh, India for their support to conduct this study.

**Conflicts of Interest:** The authors declare no conflict of interest.

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
