# Peer review of "Understanding Soil Carbon and Phosphorus Dynamics under Grass-Legume Intercropping in a Semi-Arid Region"

_agronomy, doi:10.3390/agronomy13071692_

Round 1
Reviewer 1 Report
In this study, the paper assessed the performance of Napier Bajra Hybrid (NBH)+ cowpea and trispecific hybrid (TSH) + cluster bean as compared to monocultures of NBH and TSH. It was found that the legume intensification with the forage significantly improved soil P status. It is important for enriching the mechanism of gram-bean intercropping. It is suggested that the article can be accepted after minor revision. The specific experimental data of this study is not seen in the Discussion section, and it is suggested to add the experimental data of this study, because the data is the basis for comparison with other researchers.
The quality of the English language is still very good and clearly expressed.
Author Response
Dear Reviewer,
Please find the revised version and response to your comments attached.
Thanks,
Authors

Reviewer 2 Report
ABSTRACT
Dear authors, it is not elegant to start an article by talking badly about agricultural systems.
An integrated cultivation brings countless benefits! Highlight them, start with positivity!
MM
Line 101 - What classification system was used? Citation.
Line 105 - An important data would be the field capacity of the soil, in percentage.
Line 118 - Were there phytosanitary problems? Illnesses? Fungus attack? Was any preventive application made? Insects? Pests?
RESULTS
Figure 1 could be complemented with crop evapotranspiration data, in millimeters. As a result of this information, the view of water stress due to lack of water is clear and the results easier to understand.
Figure 2 - The font is too small - difficult to read.
DISCUSSION
The vision of the article is the improvement of soil quality. However, legumes, in addition to this improvement, also serve as food. In Brazil, cowpea has a high added value and serves as a staple food in the Brazilian semi-arid regions. I believe that in the discussion the authors could explore this positive point as well.
The vision of the article is the improvement of soil quality. But one question remains: Is it feasible to implement the leguminous and grass consortium? A brief description highlighting the economic feasibility of implementing the system would enrich information!

ok
Author Response

(The authors gave the same response as above.)
